# Routing in Sparsely-gated Language Models responds to Context

**Stefan Arnold** and **Marian Fietta** and **Dilara Yesilbas**
Friedrich-Alexander-Universität Erlangen-Nürnberg
Lange Gasse 20, 90403 Nürnberg, Germany
(stefan.st.arnold, marian.fietta, dilara.yesilbas)@fau.de

## Abstract

Language Models (LMs) recently incorporate mixture-of-experts layers consisting of a router and a collection of experts to scale up their parameter count given a fixed computational budget. Building on previous efforts indicating that token-expert assignments are predominantly influenced by token identities and positions, we trace routing decisions of similarity-annotated text pairs to evaluate the context sensitivity of learned token-expert assignments. We observe that routing in encoder layers mainly depends on (semantic) associations, but contextual cues provide an additional layer of refinement. Conversely, routing in decoder layers is more variable and markedly less sensitive to context.

## 1 Introduction

*Language Models* (LMs) have demonstrated exceptional capabilities in capturing linguistic nuances (Devlin et al., 2019) and generating coherent text (Radford et al., 2019; Brown et al., 2020). However, the dense nature of their architectures, where each token is processed by the total number of parameters, inherently limits their scalability, which is considered the predominant driver for their advanced expressiveness (Kaplan et al., 2020).

Sparsely-gated *Mixture-of-Experts* (MoE) models as developed by Shazeer et al. (2017) and more recently integrated into the transformer architecture (Vaswani et al., 2017) by Lepikhin et al. (2020) and Fedus et al. (2022), emerged as a promising technique to scale up the parameter count of densely-connected language models (Brown et al., 2020). Beyond language models, this design paradigm was successfully applied to vision models (Riquelme et al., 2021) and vision-language models (Shen et al., 2023; Lin et al., 2024), showcasing its versatility and effectiveness across various tasks.

Unlike applying the same parameters to every token as in dense transformers, the guiding design principle of sparse transformers is to selectively activate a subset of parameters for each token (Bengio et al., 2013). Specifically, mixture-of-experts layers operate by incorporating routers and making them learn to dynamically direct tokens to specific parameters, referred to as *experts* (Jacobs et al., 1991). This sparsity routing addresses the scaling issues of dense transformers while maintaining a constant number of computational operations.

Since routing is central to the mixture-of-experts paradigm, most ongoing research is dedicated to identifying and relieving various challenges associated with unstable gates (Nie et al., 2021; Dai et al., 2022) and representation collapse (Chi et al., 2022; Liu et al., 2022; Do et al., 2023). Other research examined routing patterns (Zoph et al., 2022; Jiang et al., 2024; Xue et al., 2024) to assess how effectively a sparse transformer can leverage its diverse set of experts. By tracing routing decisions across expert layers, Zoph et al. (2022) discovered that expert assignments are less uniform among encoder layers than decoder layers and that meaningful specialization manifests primarily in syntactic properties rather than high-level semantics. Xue et al. (2024) further corroborated that routing is predominantly based on token identities and positions, regardless of context. This finding was termed *context-independent expert specialization* and justified by two observations: (1) tokens are routed to only a few fixed experts, and (2) consecutive token positions prefer similar experts.

**Contribution.** Given the presumption of context-independent routing, we systematically investigate the context sensitivity of *learned* token-to-expert assignments by exploiting annotated pairs of text from WordSim (Finkelstein et al., 2001), SimLex (Hill et al., 2015), SCWS (Huang et al., 2012), and WiC (Pilehvar and Camacho-Collados, 2019). We find evidence that routing is responsive to contextual cues, as words in similar contexts are more consistently assigned to the same experts compared

to words from different contexts. However, we also observe notable differences among the model components and configurations: (1) context sensitivity is more pronounced in the encoder than the decoder (in line with Zoph et al., 2022), and (2) context sensitivity increases with the total number of experts.

## 2    Background

*Mixture-of-Experts* (MoE) has a long history in machine learning, dating back to the principle of adaptive mixtures of local experts (Jacobs et al., 1991). Shazeer et al. (2017) recently introduced sparsely-gated layers by extending the mixture-of-experts paradigm with techniques for conditional computation (Bengio et al., 2013). By taking advantage of conditional computation, mixture-of-experts layers enable to scale up the number of trainable parameters while maintaining computational costs.

Building on transformer models (Vaswani et al., 2017), sparse mixture-of-experts layers can be interleaved with dense layers (Fedus et al., 2022) or upcycled from dense layers (Komatsuzaki et al., 2022). Sparse layers typically consists of a router and a fixed number of experts that are structurally identical to standard feed-forward neural networks. The router is responsible for assigning inputs to experts. Each input is projected from its hidden state to the set of experts by multiplication with the router weights, which are learned jointly with the other network parameters. To produce a gradient for the router, the output of the computation is weighted by the corresponding probability of the assignment, since this probability is differentiable. This experts-as-a-layer approach dynamically activates a fixed subset of experts, ensuring that the number of floating-point operations remain constant, regardless of the total number of experts.

To receive sufficient gradients for learning the router weights, Shazeer et al. (2017) conjectured that sparse mixture-of-experts layers require top-2 routing. As such, most implementations of sparse layers rely on two-way routing (Lepikhin et al., 2020; Du et al., 2022). However, this assumption is challenged by stable modifications for top-1 (Fedus et al., 2022; Yang et al., 2021) and adaptive top-$k$ routing (Li et al., 2023), which allows variable expert assignment based on token complexity.

To promote a balanced distribution of workload, Lepikhin et al. (2020) defined a fixed *expert capacity*, which limits the number of tokens each expert can be assigned. The expert capacity is typically specified in the form of a hyperparameter, which acts as a multiplier factor for the expected number of tokens that would be assigned to each expert under a perfect uniform distribution. If the number of tokens assigned to an expert is not enough to fill its capacity, its set of tokens is padded to fill the remaining slots. If the number of tokens assigned to an expert overflows its capacity, the extra tokens are dropped. Gale et al. (2023) addressed the token dropout issue by reformulating the computation in terms of block-sparse operations that efficiently handle the dynamism present in sparse layers.

Since routing determines the token-expert assignments and thus dictates how effectively a model can leverage its set of experts, it is of central importance for the mixture-of-experts paradigm. There are two common classes of assignment algorithms for sparse layers: *token choice* in which tokens are dispatched to top-ranked experts and *expert choice* in which experts select the top-ranked tokens.

**Token Choice.**    The most common routing strategy is *token choice* (Shazeer et al., 2017; Lepikhin et al., 2020; Fedus et al., 2022), in which routing decisions are made by greedily selecting the top-scoring experts for each token after projecting their hidden states to the number of experts.

However, the greedy nature of this routing strategy suffers from notorious load imbalance issues that may cause the routers to collapse because experts that are assigned zero tokens no longer receive gradient updates (Zhou et al., 2022). To encourage routers to make balanced token-expert assignments, additional adjustments such as *noisy gating* (Shazeer et al., 2017) and imposing an auxiliary *load balancing loss* (Fedus et al., 2022) are required. Puigcerver et al. (2024) developed a *soft routing* strategy with full differentiability that fills the capacity of experts using a weighted average of tokens. This provides a balanced and dropless mechanism for token-expert assignment.

Compared to the learning-to-route paradigm for routers (Shazeer et al., 2017; Fedus et al., 2022), an alternative strategy is to reformulate the routing algorithm as a linear assignment problem that maximizes token-expert affinity (Lewis et al., 2021; Clark et al., 2022) or to eliminate the necessity for routers: *stochastic routing* (Zuo et al., 2021) leverages a consistency regularized loss for stochastic assignment, whereas *deterministic hashing* (Roller et al., 2021) employs a parameter-free assignment algorithm that routes tokens by hashing.

**Expert Choice.** Rather than directing tokens to top-scoring experts, *expert choice* as proposed by Zhou et al. (2022) has experts independently selecting top-scoring tokens, which guarantees perfect load balancing and allows for flexible allocation.

## 3 Methodology

To illuminate the dynamics of routing with respect to context, we need to detail a sparsely-gated language model and the measurement to assess the degree of sensitivity within the sparse layers.

We employ the Switch (Fedus et al., 2022) transformer model, a sparsely-gated variant of the T5 (Raffel et al., 2020) sequence-to-sequence model, trained on a span corruption objective. This objective involves recovering variable-length contiguous segments masked in text, promoting a deeper understanding of contextual information compared to autoregressive models with dense layers (Brown et al., 2020; Touvron et al., 2023) and sparse layers (Du et al., 2022; Jiang et al., 2024). The architecture of the Switch transformer consists of an encoder and a decoder, each comprising six sparse layers that alternate between dense and sparse configurations. Each sparse layer contains a variable number of total experts in $\{8, 16, 32, 64, 128\}$, with a single active expert, where its assignment is managed through token choice routing combined with a load balancing loss. The choice of the Switch transformer model is driven by its variable configurations of experts and its simple routing strategy. By tracing token-expert assignments in the sparsely-gated layers [1], we can examine the sensitivity of the routing to similarity and the surrounding context.

**Measurements for Similarity.** To ablate whether routing is adaptive to similarity, we leverage the WordSim (Finkelstein et al., 2001) and SimLex (Hill et al., 2015) datasets. These datasets contain word pairs with human judgment on their similarity on a scale of $[0, 10]$. While WordSim captures broader relatedness in terms of associations, SimLex strictly annotates semantic similarity. For each word pair, we calculate the (layer-wise) *Jensen-Shannon Similarity* (JSS) between the routing probabilities and correlate it with the corresponding similarity annotation using the *Spearman correlation*.

**Measurements for Context.** To examine the influence of contextualization on routing decisions, we adopt the SCWS (Huang et al., 2012) dataset. Unlike WordSim and SimLex, containing word pairs in isolation, SCWS provides human judgments on the similarity of word pairs associated with a context. The inclusion of contextual cues for each word pair makes SCWS particularly suitable for measuring the extent to which context influences token-expert assignments in sparsely-gated language models. We correlate the similarity of the routing decisions for word pairs in SCWS with and without context against the provided similarity annotations.

Since most pairs of words in SCWS have dissimilar words, we further exploit the WiC (Pilehvar and Camacho-Collados, 2019) dataset [2]. Framed for binary classification, WiC is composed of a target word for which two contexts are provided that were carefully designed to trigger a specific meaning. The goal is to identify if the occurrences of the word within the contexts correspond to the same intended meaning. By comparing the routing activations separate for words from identical and different contexts, we can examine the context sensitivity of routers and identify words which are routed differently based on its contextual usage. This allows us to disentangle the effects of context from associative relationships and provide a more nuanced understanding of how routing in sparsely-gated language models is influenced by context.

## 4 Findings

To examine how consistently sparsely-gated transformers route words based on context, we calculate the similarity between the distributions of experts for word pairs and correlate them with human judgments. We interpret strong correlation coefficients as context sensitivity. Unless otherwise noted, we average the routing similarity across sparse layers.

### 4.1 Correlation with Similarity

We commence with the adaptability of routing decisions to associations in terms of relatedness and semantic similarity. Table 1 presents the correlation coefficients grouped by encoder and decoder.

For the encoder, the averaged correlation values are 0.3078 and 0.1883, respectively. These correlations indicate that the routing in sparsely-gated lan-

---

[1] We extract softmaxed router logits of word pairs. Since the Switch transformer model uses a variant of byte-pair tokenization (Kudo and Richardson, 2018), we aggregate words by mean pooling over subword components.

[2] Only 8% of the pairs of word in SCWS are identical and their assigned scores are substantially higher than those with different word pairs, *i.e.,* 6.8 compared to 3.6 on a scale from $[0, 10]$ (Pilehvar and Camacho-Collados, 2019).

Table 1: Correlation of routing probabilities with annotations of association and semantic similarity. Annotations for association were derived from `WordSim`, whereas annotations for semantic similarity were derived from `SimLex`.

| Experts | Encoder | | Decoder | |
| --- | --- | --- | --- | --- |
| | Association | Similarity | Association | Similarity |
| 8 | **0.2804** | 0.1679 | **0.0699** | 0.0510 |
| 16 | **0.3339** | 0.2070 | **0.1266** | 0.1179 |
| 32 | **0.4333** | 0.2706 | **0.1879** | 0.1127 |
| 64 | **0.3513** | 0.1485 | **0.2435** | 0.1788 |
| 128 | 0.1403 | **0.1474** | 0.0690 | **0.1317** |
| Avg. | **0.3078** | 0.1883 | **0.1394** | 0.1184 |

Table 2: Correlation of routing probabilities of word pairs with and without contextual cues to annotations of SCWS.

| Experts | Encoder | | Decoder | |
| --- | --- | --- | --- | --- |
| | w/o Context | w/ Context | w/o Context | w/ Context |
| 8 | 0.2439 | **0.3183** | 0.1497 | **0.1531** |
| 16 | 0.3493 | **0.4050** | 0.1981 | **0.2118** |
| 32 | 0.3873 | **0.4634** | 0.2997 | 0.1519 |
| 64 | 0.2562 | **0.3980** | 0.2827 | **0.3761** |
| 128 | 0.1500 | **0.3079** | 0.1382 | **0.2560** |
| Avg. | 0.2773 | **0.3785** | 0.2137 | **0.2298** |

guage models depend more on *common concepts* than by *strict meaning*, as evident by correlations for `WordSim` being consistently higher than correlations for `SimLex` across most numbers of experts. We further notice diminishing returns in routing similarities concerning the total number of experts, as evident by growing scores between 8 and 32 experts and a significant drop at 64 and 128 experts. This implies certain fluctuations (Dai et al., 2022) when a large number of experts is set.

For the decoder, the average correlation values are 0.1394 and 0.1184, respectively. Compared to routing in the encoder, the consistent yet relatively low correlations in the decoder across all configurations imply that the decoder is generally less adapted for similarity. This is particularly evident from the more modest peaks and the lack of a significant drop-off in correlation values, which indicates less pronounced expert specialization. This observation is consistent with the finding of Zoph et al. (2022) that routing is uniformly distributed.

### 4.2 Correlation with Context

We continue with the response of routing decisions to context. Table 2 presents correlation coefficients for both encoder and decoder components with and

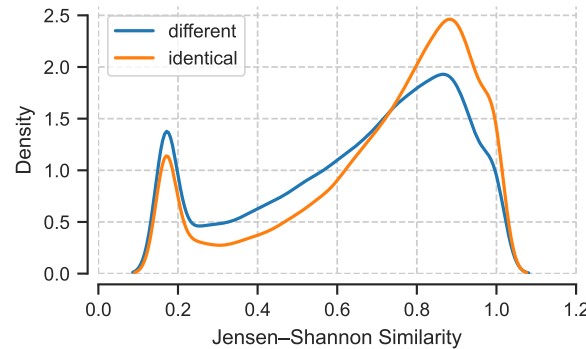

Figure 1: Density estimates for routing similarities of ambiguous words given different and identical contexts. Routing decisions are aggregated across expert configurations.

without contextual embedding.

For the encoder, the correlation coefficients without context range from 0.1500 to 0.3873, with an average value of 0.2773. This indicates a modest correlation, confirming that even without contextual cues, the routing decisions are influenced to some extent by similarity. When context is added, the correlation coefficients range from 0.3079 to 0.4634, with an average value of 0.3785. This significant increase in average correlation indicates that contextual cues enhance routing decisions, allowing the language model to capture similarities among words more effectively.

Although the average correlation in the decoder increases only slightly from of 0.2137 to 0.2298 with context, this apparent insensitivity to context is caused by notable variations in the expert configuration. With few experts, such as 8 and 16, routing decisions are hardly influenced by contextual cues. However, a larger number of experts, specifically 64 and 128, demonstrates that context can substantially inform routing decisions. This contrasts with the recent findings of Xue et al. (2024), claiming that routing in decoder layers mainly depends on token identities and positions.

Figure 1 illustrates the *Kernel Density Estimates* (KDE) for the routing similarities, distinguishing between word pairs stemming from identical contexts and those from different contexts of `WiC`. Note that the density estimates are calculated across expert configurations in $\{8, 16, 32, 64, 128\}$.

The density curves are shaped similarly with a bimodal distribution, with density peaks at low and high values for the routing similarities visibly distinguishable. The density peak at high values indicates that, for many word pairs in identical contexts, the routing probabilities are quite simi-

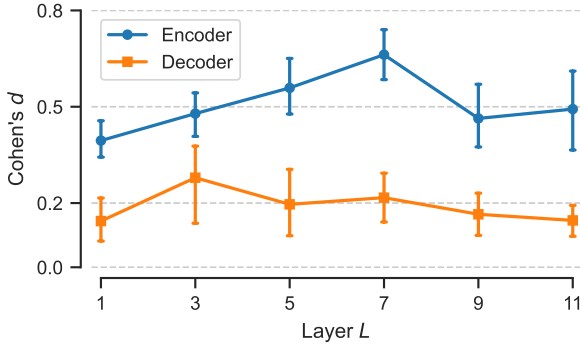

Figure 2: Layer-wise effect sizes using Cohen's $d$ on the routing similarities of ambiguous words given some context. Routing decisions are aggregated across expert configurations.

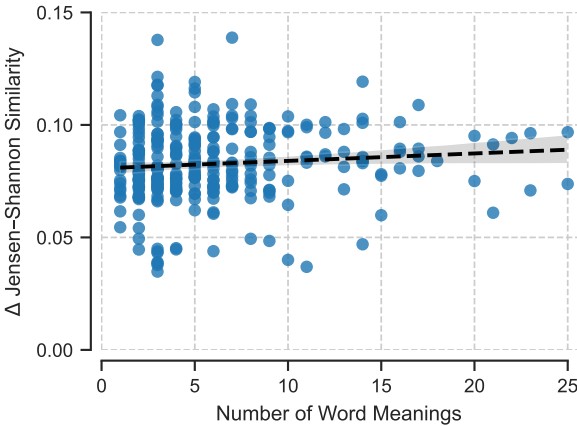

Figure 3: Differences in routing similarities for a set of ambiguous words given some context, as a function of the number of unique meanings derived from WordNet.

lar, reflecting a commendable level of consistency in routing. The density peak at lower values suggests diverse routing patterns for many word pairs from different contexts, as desired when the context differs significantly. However, the overlap in the density curves implies that some word pairs receive similar routing despite having dissimilar meanings, which may occur in texts where the contexts are not substantially distinct, or the context differences are not clearly delineated by the language model.

Figure 2 provides a layered investigation of the effect sizes of context sensitivity in the encoder and decoder layers. We measured the effect size using Cohen's $d$ by comparing the difference in routing similarities of words from identical and different contexts of WiC. We find that context is consistently significant for the routers in the encoder layers, whereas the routers in the decoder layers maintain a relatively stable and considerably lower effect sizes to context. Specifically, context integrates progressively in the early layers, peaks in the middle layers, and then slightly diminishes in the rear layers. This pattern can be attributed to late routers being specialized for span reconstruction.

### 4.3 Correlation with Ambiguity

Since words can have multiple, potentially unrelated, meanings depending on the context, we are interested if routing decisions for ambiguous words vary with the number of meanings. Figure 3 plots differences in routing similarities against the number of word meanings derived from WordNet (Miller, 1995) [3]. Although the trend line indicates that the context sensitivity of words correlates (in-

---

[3]WordNet provides sets of synonyms that share a common meaning. To measure the number of meanings of a word, we counted the occurrence of a word in distinct *synsets*.

significantly) with the number of distinct meanings, there is considerable variability, particularly for words with few meanings. This variability suggests that factors besides the number of meanings, such as word frequency, may determine the consistency of token-expert assignments in learned routers.

## 5 Conclusion

Given the claims surrounding the factors influencing routing decisions in sparsely-gated mixture-of-experts language models (Zoph et al., 2022; Xue et al., 2024), we provide valuable insights into the influence of similarity and context. While similarity, encapsulated by token identities, form a stable basis for routing decisions, contextual cues provide an additional layer of refinement. However, the varying impact of context on the encoder and decoder reveals different sensitivities within the model components. The encoder demonstrates a strong ability to assign words in similar contexts consistently, revealing a high sensitivity to contextual cues, especially for configurations with many experts per sparse layer. The response of the decoder to context is poorer and more variable. This variability indicates instabilities in the utilization of context with respect to the number of experts.

Since our study demonstrates that context plays a significant role in routing, we hope that our approach sparks research on other linguistic properties and their influence on routing decisions, *e.g.*, the influence of (affixal) negation (van Son et al., 2016) or the consistency of routing for multi-word expressions (Kochmar et al., 2020).

**Limitation.** Challenging current claims about the context sensitivity of sparsely-gated language models, this study is limited by its focus on the `Switch` transformer model with its encoder-decoder architecture. Therefore, our findings may not be directly applicable to other types of transformer architectures, such as purely autoregressive models optimized with next-word prediction. We thus advocate for endeavors that expand the scope of analysis to cover a broader range of transformer architectures and develop more refined routing mechanisms to better integrate contextual cues, particularly for words with high polysemy.

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
