# OpenReview forum: "Routing in Sparsely-gated Language Models responds to Context"
_EMNLP/2024/Workshop/BlackBoxNLP — BlackboxNLP 2024_

### Official Review · Reviewer_xmUr · 2024-08-31

**Overall Assessment:** 4
**Confidence:** 2

**Best Paper:**

1

**Best Paper Justification:**

-

**Comments Questions Suggestions And Typos:**

Typos

1. line 095 should be conditional computation.

**Paper Summary:**

The study investigates how context influences the assignment of tokens to experts in sparsely gated expert layers. Using annotated text pairs from various datasets, the work demonstrates that routing is sensitive to contextual cues, meaning that words in similar contexts are often assigned to the same experts. However, they note differences depending on the model components and configurations: context sensitivity is stronger in the encoder than in the decoder and increases with the number of experts.

**Summary Of Strengths:**

1. The paper is written clearly with good illustrations and sufficient background depth.
2. The paper takes a simple setting to dig into understanding the mechanistic aspects of an important class of models which is well aligned with the theme of this workshop.
3. The paper sheds light on how token-to-expert assignments in a model are influenced by context and similarity.

**Summary Of Weaknesses:**

1. The paper's findings are weakened by the narrow selection of model and data to run its studies. I am of the opinion that a deep investigation in simple settings is important but it does not have to preclude subsequently expanding scope.
2. It would be good to put some of the numerical results regarding the correlations into perspective by using qualitative examples, interesting failure modes where context did not help route to similar experts etc.

---

### Official Review · Reviewer_seU3 · 2024-09-06

**Overall Assessment:** 4
**Confidence:** 4

**Best Paper:**

1

**Best Paper Justification:**

-

**Comments Questions Suggestions And Typos:**

- Line 152: Typo (choic)
- Figure 3: "The marked trend line indicates that the context sensitivity of words correlates with the number of distinct meanings" - the correlation seems very small, is it statistically significant?

**Paper Summary:**

The paper analyses the context-sensitivity of routing in a sparsely-gated transformer model (Switch). Contrary to previous research, which found that routing is mostly based on token identity, the authors find that routing is in fact context-sensitive: words in similar contexts are more often assigned to the same experts compared to words in different contexts.

**Summary Of Strengths:**

- Well-written paper
- Good and elaborate background section
- Well-motivated experimental design
- Separate analyses on the encoder and decoder (Table 1 and 2), as well as on individual layers (Figure 2), which is valuable to the interpretability community

**Summary Of Weaknesses:**

I would say that the motivation for doing this research is left somewhat implicit - the contributions are stated clearly and the results help us better understand the dynamics of routing, but it is not necessarily clear how these contributions may be used in future research, or how they help advance the interpretability field general.

---

### Official Review · Reviewer_JAkj · 2024-09-09

**Overall Assessment:** 5
**Confidence:** 2

**Best Paper:**

2

**Best Paper Justification:**

Results and methods in this paper can be used as a starting point of a lot of interesting future work

**Comments Questions Suggestions And Typos:**

l. 095 'By taking advantage of computational computation' --> 'By taking advantage of conditional computation'
l. 152 'token choic' --> 'token choice'
l. 273 'common common concepts' --> 'common concepts'?

**Paper Summary:**

The paper studies factors behind routing decisions in a MoE model (Switch). More specifically, the authors test whether similarity by association affects routing; the same for a more strict notion of semantic similarity, and for similarity-in-context. The comparison is made between encoder and decoder parts of the model. The conclusions are mostly consistent with existing observations about encoder layers being more sensitive to the semantics of tokens when it comes to routing, with decoder layers showing a less clear pattern. Context-sensitivity of routing decisions in encoder layers is a new fact that points even stronger towards the semantic basis of routing.

**Summary Of Strengths:**

It is an exciting paper that is written well and explores under-explored (because relatively new!) components of recent language models. The results are very promising, and the method seems solid. I would also like to note creative use of existing datasets for the purpose of the study.

I really like the suggestions made in the conclusion that this approach could be applied to systematic studies of various linguistic phenomena. I would love to see this type of work!

**Summary Of Weaknesses:**

I really don't have any deep complaints, the paper is great. Maybe I missed something because I haven't thought about MoE models in much depth -- I reflect this fact in my confidence score.

---

### Decision · Program_Chairs · 2024-09-19

**Decision:**

Accept

**Comment:**

The reviewers unanimously agreed that the paper is well written and the approaches taken are generally solid. Overall, this paper presents an exciting new direction and opens up many promising possibilities of future investigation.